# Anti-Melanogenic Properties of Velutin and Its Analogs [note 1]

**DOI:** 10.3390/molecules26103033

**Published:** 2021-05-19

**Authors:** Se-Hui Jung, Hee-Young Heo, Jung-Won Choe, Jaehyun Kim, Kooyeon Lee

**Affiliations:** 1Department of Bio-Health Technology, Division of Biomedical Convergence, College of Biomedical Science, Kangwon National University, Chuncheon 24341, Korea; 96-shjung@hanmail.net (S.-H.J.); gmldud940315@naver.com (H.-Y.H.); cjh9531@naver.com (J.-W.C.); 1342113@naver.com (J.K.); 2Research Institute, K-medichem Co., Ltd., Chuncheon 24341, Korea

**Keywords:** velutin derivatives, melanin synthesis, tyrosinase activity, SAR study

## Abstract

Velutin, one of the flavones contained in natural plants, has various beneficial activities, such as skin whitening, as well as anti-inflammatory, anti-allergic, antioxidant, and antimicrobial activities. However, the relationship between the structure of velutin and its anti-melanogenesis activity is not yet investigated. In this study, we obtained 12 velutin derivatives substituted at C5, C7, C3′, and C4′ of the flavone backbone with hydrogen, hydroxyl, and methoxy functionalities by chemical synthesis, to perform SAR analysis of velutin structural analogues. The SAR study revealed that the substitution of functional groups at C5, C7, C3′, and C4′ of the flavone backbone affects biological activities related to melanin synthesis. The coexistence of hydroxyl and methoxy at the C5 and C7 position is essential for inhibiting tyrosinase activity. However, 1,2-diol compounds substituted at C3′ and C4′ of flavone backbone induce apoptosis of melanoma cells. Further, substitution at C3′ and C4′ with methoxy or hydrogen is essential for inhibiting melanogenesis. Thus, this study would be helpful for the development of natural-derived functional materials to regulate melanin synthesis.

## 1. Introduction

Flavones, a type of flavonoid consisting of two phenyl rings and a benzopyran functionality, are widely present in natural plants, including vegetables and fruits [1]. In many compounds, the flavone scaffolds are considered important core structures that act on various targets molecules via simple modifications. In particular, flavone derivatives with metabolic or synthetic modification are known to have anti-inflammatory, antiestrogenic, anti-allergic, antioxidant, antimicrobial, antitumor, and anti-proliferation activities [2,3,4]. Furthermore, simple substitution of the flavone backbone with functional groups like hydroxyl and methoxy, results in very different biological activities. Due to the broad biological activities of flavones, their structure–activity relationship (SAR) attracted interest in the field of medicinal chemistry, which helped to discover some lead compounds to alleviate numerous diseases. For example, the previous SAR study revealed that, as the number of hydroxyl groups increased, the radical scavenging effect of flavones increased [5].

Melanin is a natural pigment that is involved in the determination of skin, eye, and hair color, and has an important role in the protection of skin from damage by harmful light, such as ultraviolet rays [6,7,8]. Despite its beneficial effects, abnormal regulation of melanin synthesis is responsible for pigmentary disorders that include albinism, vitiligo, melasma, freckles, and lentigo [9,10]. Melanin is synthesized in melanocytes, which are specialized cells located at the basal layer of the epidermis, through multistep catalytic reactions of enzymes such as tyrosinase, tyrosinase-related protein 1, and tyrosinase-related protein 2 [9,11]. Tyrosinase plays a key role in the two rate-limiting reactions of melanin synthesis, by the hydroxylation of l-tyrosine to l-3,4-dihydroxyphenylalanine (l-DOPA), and the oxidation of l-DOPA to dopaquinone [12]. Thus, various regulators of the tyrosinase activity were studied for treating pigmentary disorders [13]. Whitening reagents based on tyrosinase inhibition, such as hydroquinone, azelaic acid, arbutin, and kojic acid, were reported to prevent skin hyperpigmentation [11,14]. However, due to their cellular toxicity, low efficacy, and low stability in the presence of oxygen and water, whitening ingredients from natural products are considered in the cosmetic research and development field to be an alternative strategy to prevent hyperpigmentation.

Velutin belongs to a class of flavonoids that have hydroxyl, methoxy, methoxy, and hydroxyl groups at C5, C7, C3′, and C4′ of the flavone backbone, respectively (Figure 1). Velutin is found in the pulp of acai fruit, and is known to exhibit anti-inflammatory and antioxidant activities [15,16]. Furthermore, velutin obtained by the deglycosylation of homoflavoyadorinin B in mistletoe extract has an inhibitory effect on tyrosinase and melanogenesis [8]. However, the relationship between the structural modification of velutin and its anti-tyrosinase and anti-melanogenesis activities are not yet investigated.

In this study, we prepared 12 velutin derivatives by substitution at C5, C7, C3′, and C4′ of the flavone backbone with hydrogen, hydroxyl, and methoxy functionalities, to perform SAR analysis of the velutin structural analogues. All 12 derivatives were evaluated for their inhibitory activity against tyrosinase activity and melanogenesis. We found that the coexistence of the methoxy group at the C7 position and hydroxyl groups at the C5 and C4′ positions in the flavone skeleton was essential to have inhibitory activity. In addition, the 1,2-diol form with hydroxyl groups at the C3′and C4′ positions of the flavone induced apoptosis in the melanoma cells. Thus, these findings would be helpful for investigating the effect of flavone derivatives on the biosynthesis of melanin, and for developing new chemical drugs to suppress melanin synthesis.

## 2. Results and Discussion

### 2.1. Chemical Synthesis of Velutin Derivatives

In order to understand the relationship between the structural modification of the velutin backbone and its biological activities, eleven velutin derivatives (**V1–V12**, except **V11**) were chemically synthesized from 2-hydroxy acetophenone derivatives and aromatic aldehyde derivatives, via five-step conventional methods [17]. Three velutin derivatives, **V1** (velutin), **V5**, and **V8**, were synthesized via a seven-step reaction, as shown in Scheme 1. First, to synthesize 1-(2-(benzyloxy)-6-hydroxy-4-methoxyphenyl)ethan-1-one (**V1e**), bis-MOM-protected acetophenone (**V1b**) was synthesized from commercially available 2′,4′,6′-trihydroxyacetophenone (**V1a**), subjected to deprotection reaction, after synthesis of benzylated acetophenone (**V1c**), and selective substitution of the methoxy group at the C4 position of 1-(2-(benzyloxy)-4,6-dihydroxyphenyl)ethan-1-one (**V1d**) was performed. Then, **V1**, **V5**, and **V8** derivatives were synthesized by the subsequent aldol condensation, oxidative cyclization, and reduction from the modified acetophenone (**V1e**).

**V2** and **V3** were synthesized via subsequent reactions, as shown in Scheme 2; *O*-alkylation at R2 and R1 of the commercially available two dihydroxyacetophnone compounds, aldol condensation, oxidative cyclization, and deprotection.

Four derivatives, including **V4**, **V6**, **V9**, and **V10** were synthesized via subsequent reactions, as shown in Scheme 3; *O*-alkylation at C4 and C6 of the commercially available 2′,4′,6′-trihydroxyacetophnone compound (V1a), aldol condensation, oxidative cyclization, and deprotection.

The V7 derivative was synthesized via subsequent reactions, as shown in Scheme 4; benzyl protection at C4 and C6 of the commercially available 2′,4′,6′-trihydroxyacetophnone compound (**V1a**), aldol condensation, oxidative cyclization, and reduction.

**V12** derivative was synthesized via a two-step reaction, as previously reported [18]; aldol condensation of two commercially available 2′-hydroxyacetophnone and benzaldehyde, and oxidative cyclization.

Table 1 summarizes the substitution group and the overall yield of the synthesized velutin derivatives.

### 2.2. Characterization of Synthetic Velutin Derivatives

#### 2.2.1. Synthesis of Velutin (**V1**)

(**V1**): ^1^H NMR (400 Mhz, DMSO-d_6_) *δ* (12.97 (s, 1H), 10.01 (br, s, 1H), (7.61–7.59) (m, 2H), (6.96–6.94) (m, 2H), 6.80 (d, *J* = 1.52 Hz, 1H), 6.37 (d, *J* = 2.08 Hz, 1H), 3.91 (s, 3H), 3.88 (s, 3H)).

#### 2.2.2. Synthesis of 2-(4-Hydroxy-3-methoxyphenyl)-7-methoxy-4*H*-chromen-4-one (**V2**)

(**V2**): ^1^H NMR (400 Mhz, DMSO-d_6_) *δ* (9.88 (s, 1H), 7.92 (d, *J* = 8.84 Hz, 1H), (7.61–7.58) (m, 2H), 7.32 (d, *J* = 2.36 Hz, 1H), 7.05 (dd, *J*_1_ = 8.80 Hz, *J*_2_ = 2.40 Hz, 1H), (6.95–6.93) (m, 1H), 6.89 (s, 1H), 3.92 (s, 3H), 3.91 (s, 3H)).

#### 2.2.3. Synthesis of 5-Hydroxy-2-(4-hydroxy-3-methoxyphenyl)-4*H*-chromen-4-one (**V3**)

(**V3**): ^1^H NMR (400 Mhz, DMSO-d_6_) *δ* (12.85 (s, 1H), (7.69–7.62) (m, 3H), 7.21 (d, *J* = 8.36 Hz, 1H), 7.07 (s, 1H), 6.96 (d, *J* = 8.16 Hz, 1H), 6.81 (d, *J* = 8.16 Hz, 1H), 3.91 (s, 3H)).

#### 2.2.4. Synthesis of Genkwanin (**V4**)

(**V4**): ^1^H NMR (400 Mhz, DMSO-d_6_) *δ* (12.96 (s, 1H), (7.97–7.95) (m, 2H), (6.95–6.92) (m, 2H), 6.85 (s, 1H), 6.77 (d, *J* = 2.24 Hz, 1H), 6.38 (d, *J* = 2.28 Hz, 1H), 3.87 (s, 3H)).

#### 2.2.5. Synthesis of 5-Hydroxy-7-methoxy-2-(3-methoxyphenyl)-4*H*-chromen-4-one (**V5**)

(**V5**): ^1^H NMR (400 Mhz, CDCl_3_) *δ* (12.71 (s, 1H), (7.45–7.39) (m, 3H), (7.10–7.07) (m, 1H), 6.65 (s, 1H), 6.50 (d, *J* = 2.28 Hz, 1H), 6.38 (d, *J* = 2.24 Hz, 1H), 3.90 (s, 3H), 3.89 (s, 3H)).

#### 2.2.6. Synthesis of 2-(4-Hydroxy-3-methoxyphenyl)-5,7-dimethoxy-4*H*-chromen-4-one (**V6**)

(**V6**): ^1^H NMR (400 Mhz, DMSO-d_6_) *δ* (7.53–7.52) (m, 2H), (6.93–6.86) (m, 1H), 6.86 (d, *J* = 2.24 Hz, 1H), 6.49 (d, *J* = 2.20 Hz, 1H), 3.90 (s, 3H), 3.89 (s, 3H), 3.83 (s, 3H)).

#### 2.2.7. Synthesis of Chrysoeriol (**V7**)

(**V7**): ^1^H NMR (400 Mhz, DMSO-d_6_) *δ* (12.97 (s, 1H), (7.57–7.55) (m, 1H), 6.93 (d, *J* = 8.92 Hz, 1H), 6.88 (s, 1H), 6.47 (d, *J* = 1.87 Hz, 1H), 6.16 (d, *J* = 1.91 Hz, 1H), 3.89 (s, 3H)).

#### 2.2.8. Synthesis of 2-(3,4-Dihydroxyphenyl)-5-hydroxy-7-methoxy-4*H*-chromen-4-one (**V8**)

(**V8**): ^1^H NMR (400 Mhz, DMSO-d_6_) *δ* (12.99 (br, s, 1H), (7.47–7.44) (m, 2H), 6.90 (d, *J* = 8.21 Hz, 1H), (6.74–6.73) (m, 2H), 6.38 (d, *J* = 2.24 Hz, 1H), 3.88 (s, 3H)).

#### 2.2.9. Synthesis of 2-(3,4-Dimethoxyphenyl)-5-hydroxy-7-methoxy-4*H*-chromen-4-one (**V9**)

(**V9**): ^1^H NMR (400 Mhz, CDCl_3_) *δ* (12.80 (s, 1H), 7.53 (dd, *J*_1_ = 8.48 Hz, *J*_2_ = 2.12 Hz, 1H), 7.34 (d, *J* = 2.08 Hz, 1H), 6.98 (d, *J* = 8.54 Hz, 1H), 6.50 (d, *J* = 2.24 Hz, 1H), 6.37 (d, *J* = 2.22 Hz, 1H), 3.98 (s, 3H), 3.97 (s, 3H), 3.89 (s, 3H)).

#### 2.2.10. Synthesis of 2-(3,4-Dimethoxyphenyl)-5,7-dimethoxy-4*H*-chromen-4-one (**V10**)

(**V10**): ^1^H NMR (400 Mhz, DMSO-d_6_) *δ* (7.64 (dd, *J*_1_ = 8.48 Hz, *J*_2_ = 2.16 Hz, 1H), 7.53 (d, *J* = 2.12, 1H), 7.11 (d, *J* = 8.62, 1H), 6.87 (d, *J* = 2.30, 1H), 6.77 (s, 1H), 6.50 (d, *J* = 2.30, 1H), 3.91 (s, 3H), 3.88 (s, 3H), 3.85 (s, 3H), 3.83 (s, 3H)).

#### 2.2.11. 2-Phenylchromen-4-one (**V12**)

(**V12**): ^1^H NMR (400 Mhz, CDCl_3_) *δ* (8.24 (dd, 1H, *J*_1_ = 1.66 Hz, *J*_2_ = 7.94 Hz), (7.96–7.93) (m, 2H), (7.73–7.69) (m, 1H), (7.59–7.57) (m, 1H), (7.56–7.53) (m, 3H), 7.45–7.41 (m, 1H), 6.84 (s, 1H)).

### 2.3. Effect of Velutin Derivatives on ABTS Radical Scavenging Activity

The ABTS radical scavenging activity of the twelve velutin derivatives or arbutin (positive control) was then evaluated in the concentration range 50–200 μM, and Table 2 lists their half-maximal effective concentration (EC_50_) values. Most derivatives, except **V10**, showed dose-dependent radical scavenging activity (data not shown), with EC_50_ values ranging (5.47–100.13) μM (Table 2). In particular, **V8** and **V11** (luteolin) compounds with a 1,2-diol-type structure, showed a better radical scavenging activity than the other derivatives. In contrast, the **V10** or **V12** compounds substituted with methoxy or hydrogen group at C5, C7, C3′, and C4′ of the backbone, had no significant effect on radical scavenging activity. These results indicated that the presence of hydroxyl group on the velutin backbone was critical for its radical scavenging activity. Furthermore, increase in the number of methoxy functionality at C5, C7, C3′, and C4′ of the backbone lowered ABTS radical scavenging activity.

### 2.4. Effect of Velutin Derivatives on Mushroom Tyrosinase Inhibitory Activity

To further understand the effect of the electrostatic potential of the derivatives on their inhibitory activities against tyrosinase, the inhibitory effect of the twelve velutin derivatives on mushroom tyrosinase was evaluated. The five velutin derivatives **V1**, **V4**, **V8**, **V9**, and **V11** had potential inhibitory activity against mushroom tyrosinase, with IC_50_ values ranging 37–910 µM, but the other seven velutin derivatives **V2**, **V3**, **V5**, **V6**, **V9**, **V10**, and **V12** had no significant effect (Table 2). These results indicated that the number of hydroxyl groups at C5, C7, C3′, and C4′ of the flavone backbone was an important factor for their inhibitory activity against tyrosinase. In particular, the **V11** compound showed the lowest IC_50_ value in the twelve derivatives, while **V10** and **V12** had no significant effect on tyrosinase activity, suggesting that the existence of hydroxyl group, rather than methoxy or hydrogen group, at C5, C7, C3′, and C4′ of the flavone backbone was more important for its anti-tyrosinase effect. Furthermore, hydroxyl and methoxy functionalities at the R^1^ and R^2^ positions might play an important role in the inhibition of tyrosinase.

### 2.5. Cytotoxicity of Velutin Derivatives in Melanoma Cells

Cytotoxicity is a concern for any potential compound, and thus after treating the cells with derivatives, we quantitatively measured the cytotoxicity of the eleven velutin derivatives that had the potential to inhibit melanogenesis, except for **V12**, which had no potential to inhibit melanogenesis through MTT assay. Nine derivatives, excluding **V8**, **V10**, and **V11**, had no significant effect on cell viability at concentrations up to 20 μM (data not shown). However, the treatment with **V8**, **V10**, and **V11**, induced cell death in a concentration-dependent manner, with death rates of 71.8, 24.0, and 69.7%, respectively, at 20 μM (Figure 2). This result indicates that the 1,2-diol form with hydroxyl groups at the C3′ and C4′ positions of the flavone induced apoptosis in melanoma cells, which was consistent with the previous report explaining the cytotoxicity of flavonoid derivatives in 1,2-diol form, such as catechin and epicatechin [19].

### 2.6. Effect of Velutin Derivatives on Melanogenesis in Melanoma Cells

Next, the effects of the nine velutin derivatives excluding **V8** and **V11**, which have cytotoxicity, on melanogenesis stimulated by α-melanocyte stimulating hormone (α–MSH) in melanoma cells were determined. Treatment with α–MSH stimulated a 1.9-fold increase of the melanin content of the melanoma cells (Figure 3A). We found that α–MSH-stimulated melanogenesis was completely inhibited by **V1** and **V2** (*p* < 0.001), and partially inhibited by **V3** (50.7%, *p* < 0.005), **V4** (69.1%, *p* < 0.001), and **V5** (39.3%, *p* < 0.005), while α–MSH-stimulated melanogenesis was not significantly changed by **V7**, **V9**, and **V10** (Figure 3A). Surprisingly, the **V6** derivative significantly increased melanin synthesis by 3.0-fold (*p* < 0.001). These results suggest that the velutin derivatives with similar structures but different substitution groups, have different biological activity on melanogenesis. Further, the inhibitory activities of the nine velutin derivatives on α–MSH-mediated intracellular tyrosinase activation were determined, since this enzyme plays a key role in melanogenesis. Treatment with α–MSH induced an approximately 1.7-fold activation of tyrosinase in B16F10 cells (Figure 3B). Four derivatives **V1** (87%, *p* < 0.001), **V3** (97%, *p* < 0.001), **V4** (100%, *p* < 0.001), and **V5** (80%, *p* < 0.001) reversed tyrosinase activation mediated by α–MSH, whereas four other derivatives **V2**, **V7**, **V9**, and **V10** did not (Figure 3B). These results indicate that the four derivatives **V1**, **V3**, **V4**, and **V5** inhibited α–MSH-induced melanin synthesis, by inhibiting the intracellular tyrosinase activity in B16F10 cells.

### 2.7. In Silico Molecular Docking Simulation of Enzyme Inhibition

To predict the interaction between velutin derivatives and tyrosinase, an in silico molecular docking study was performed using the Maestro software. The molecular docking study revealed that **V9** and **V10** derivatives, highly substituted with methoxy groups, did not bind to the enzyme (Table 3); this result was consistent with our results obtained from the in vitro experiment. Further, most of the flavone derivatives, except **V9** and **V10**, interact with tyrosinase with binding energies ranging from −4.983 to −5.677 kcal/mol (Table 3). The two derivatives, **V1** and **V4**, showed inhibitory activity against tyrosinase activity in melanoma cells, with docking scores of −5.043 and −5.136 kcal/mol, respectively. They were stabilized by π–π stacking with Phe 264 (A and C ring) and His 259 (B ring), and π–cation interaction with Arg 268 (A ring), as shown in Figure 4. Two derivatives that were highly substituted with hydroxyl groups, **V7** and **V11**, formed hydrogen bonds with enzyme, and showed relatively high docking scores of −5.174 and −5.677 kcal/mol, respectively. This result was consistent with our results, showing relatively low IC_50_ values against tyrosinase activity obtained from in vitro experiment. However, **V7** had no effect on melanogenesis and tyrosinase activity in melanoma cells, while **V11** showed cytotoxicity in melanoma cells.

## 3. Experimental Section

### 3.1. Chemistry

Commercially available reagents were used without additional purification. All reaction mixtures were magnetically stirred, and were monitored by thin-layer chromatography, using silica gel pre-coated glass plates visualized with UV light, and then developed using either iodine or a solution of anisaldehyde. Flash column chromatography was carried out using silica gel ((230–400) mesh). ^1^H NMR (400 Mhz) and spectra were recorded by NMR spectrometry. Deuterated chloroform was used as the solvent and chemical shift values (*δ*) were reported in parts per million, relative to the residual signals of this solvent [*δ* 7.26 for ^1^H (chloroform-*d*), *δ* 2.50 for ^1^H (dimethyl sulfoxide-*d*)].

### 3.2. Materials

Reagents such as 2,2′-azinobis(3-ethylbenzothiazoline-6-sulfonic acid (ABTS) and 3-(4,5-dimethylthiazol-2-yl)-2,5-diphenyltetrazolium bromide (MTT), α-melanocyte stimulating hormone (*a*–MSH), and enzymes, such as mushroom tyrosinase, were obtained from Sigma–Aldrich (St. Louis, MO, USA). Potassium persulfate (K_2_S_2_O_8_), pyrocatechol violet, and 3,4-dihydroxy-L-phenylalanine (L-DOPA) were purchased from Alfa Aesar (Haverhill, MA, USA).

### 3.3. Determination of ABTS Free Radical Scavenging Activity

In order to evaluate the antioxidant activity of the velutin derivatives, the ABTS radical scavenging activity was determined, as previously described [20]. To generate the ABTS radical, 10 mL of 7 mM ABTS was mixed with 176 μL of 140 mM potassium peroxydisulfate in dH_2_O, and incubated in the dark at room temperature (RT) for 16 h, prior to use. The ABTS radical solution was diluted with absolute methanol, to obtain an absorbance of near 0.7 at 734 nm. Aliquots of 100 μL of each derivative in the indicated concentration range 2 to 200 μM were added to 100 μL of the diluted ABTS radical solution, and incubated for 10 min in the dark at RT. The absorbance was then measured at 732 nm using a SpectraMax M5 Multi-Mode microplate reader (Molecular Devices, Sunnyvale, CA, USA). The ABTS radical scavenging activity was calculated as follows:ABTS radical scavenging activity (%) = 1 − (Asample/Acontrol) × 100(1)

### 3.4. In Vitro Mushroom Tyrosinase Inhibition

The inhibitory effect of velutin derivatives on tyrosinase activity was assessed by the amount of dopachrome synthesized from the catalytic reaction of tyrosinase [8,21]. In brief, 50 μL of velutin derivatives in the concentration range of 0.01–1 mM was mixed with 50 μL of 50 U/mL mushroom tyrosinase, in 50 mM phosphate buffered saline (PBS; 8.1 mM Na_2_HPO_4_, 1.2 mM KH_2_PO_4_, pH 6.8, 2.7 mM KCl, 138 mM NaCl) in a 96-well plate, and incubated for 30 min at RT. Then, 100 μL of 0.4 mM L-tyrosine was added to each well, followed by incubation for an additional 10 min at 37 °C. The absorbance of the resulting solution was measured at 475 nm, using a SpectraMax M5 Multi-Mode microplate reader.

### 3.5. Cell Culture

B16F10 murine melanoma cells were cultured in DMEM medium (Gibco, Gaithersburg, USA) supplemented with 10% heat-inactivated fetal bovine serum (Gibco), 100 units/mL penicillin, and 100 μg/mL streptomycin (Gibco), at 37 °C under humidified 5% CO_2_.

### 3.6. Cell Viability

The viability of B16F10 cells was determined using an MTT assay, as previously described [22]. In brief, B16F10 cells were seeded in 24-well plates at a density of 1 × 10^4^ cells per well. After 24 h, the cells were treated with the indicated concentrations of velutin derivatives for 48 h. The cells were then incubated with MTT solution for 4 h, and the reduced formazan crystals were dissolved in DMSO. The resulting solution was transferred to 96-well plates, and the absorbance was measured at 540 nm, using a SpectraMax M5 Multi-Mode microplate reader.

### 3.7. Melanin Content Determination

The melanin content was determined as previously described, with some modifications [23]. The melanoma cells were cultured in a 6-well plate for 24 h. They were treated with the indicated concentrations of velutin derivatives for a further 48 h, in the presence of 100 nM α–MSH. After washing twice with chilled Dulbecco’s phosphate buffered saline supplemented with calcium chloride and magnesium chloride (D-PBS, Gibco), the resulting cells were detached by incubation with trypsin–EDTA solution. After centrifugation at 1000 rpm for 3 min, the cell pellet was dissolved in 150 μL of 1 M NaOH containing 10% DMSO for 1 h at 60 °C. The melanin content was determined by the absorbance at 405 nm, using the microplate reader.

### 3.8. Determination of Cellular Tryosinase Activity in Melanoma Cells

Tyrosinase activity in B16 cells was examined on the basis of the amount of dopachrome produced from the catalytic reaction of intracellular tyrosinase [24]. In brief, the melanoma cells were cultured in a 6-well plate for 24 h, followed by treatment with different concentrations of velutin derivatives for a further 48 h, in the presence of 100 nM of α–MSH. After washing twice with ice-cold D-PBS, the cells were lysed in 200 μL of radio-immunoprecipitation assay (RIPA) buffer (Sigma-Aldrich), containing protease and phosphatase inhibitors. After centrifugation of the cell lysate collected from each well at 15,000× *g* for 15 min, 100 μL of supernatant was mixed with 100 μL of 1 mM L-DOPA in PBS (pH 6.8), followed by incubation for 30 min at 37 °C. The absorbance of dopachrome was measured at 475 nm using the microplate reader. Data were normalized with protein concentration, determined by bicinchoninic acid assay.

### 3.9. Molecular Modeling

The crystal structure of mushroom tyrosinase for molecular modeling was an Agaricus bisporus tyrosinase (PDB dose: 2Y9X) obtained from the Protein Data Bank (PDB). The enzyme was prepared by using the protein wizard preparation workflow embedded in the Maestro program (Maestro, version 11.9.011, Schrödinger, LLC, New York, NY, USA, 2019). Water, and all the other molecules present in the pdb files, were removed. Molecular docking was performed using the Induced Fit Docking (IFD) protocol (Schrödinger Suite 2019 Induced Fit Docking protocol), as previously reported [25].

### 3.10. Statistical Analysis

All data in this study were expressed as the mean ± standard deviation (SD) from three independent experiments. Statistical analyses were performed using the Graph-Pad Prism 8.0 (GraphPad Software Inc., La Jolla, CA, USA). The differences between the mean values of the control and the exposed groups were analyzed using one-way analysis of variance (ANOVA). A value of *p* < 0.05 was considered to be statistically significant.

## 4. Conclusions

In this study, we obtained twelve velutin derivatives through chemical synthesis, and comparatively analyzed the relationship between the chemical structure of these derivatives and their activity related to melanin synthesis. The SAR study revealed that the substitution of functional groups at C5, C7, C3′, and C4′ of the flavone backbone affected the biological activities related to melanin synthesis (Figure 5). The coexistence of hydroxyl and methoxy at R^1^ and R^2^, respectively, was essential for inhibiting tyrosinase activity. However, the 1,2-diol compounds substituted at R^3^ and R^4^ induced apoptosis of melanoma cells. Further, substitution at R^3^ and R^4^ with methoxy or hydrogen was essential for inhibiting melanogenesis. This study would be helpful for the development of natural-derived functional materials, to regulate melanin synthesis.

## Data Availability

The data presented in this study are available on request from the corresponding author.

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
