# Peer review of "Anti-Melanogenic Properties of Velutin and Its Analogs"

_molecules, 2021, doi:10.3390/molecules26103033_

Round 1

Reviewer 1 Report

In this paper the authors describe the anti-melanogenic properties of velutine and some flavone analogs, in order to assess the SARs. This paper is certainly interesting, due to the significative role of flavone derivatives in a number of medicinal chemistry fields, but in my opinion the aims of this work should be expressed more clearly.  A series of hydroxy/methoxy flavone derivatives is reported, obtained by reacting three 2-hydroxy acetophenone derivatives with three properly substituted aldehydes. The velutin structure is missed, and the reason on the basis of the substituents selection is not explained. Regarding the chemistry of compounds, scheme 1 reports no data on the reagents and conditions used, and the nature of the R’ substituent on the aldehyde is unclear (benzyl? Why? When and how is it removed?). In fact, no synthetic procedure is defined, even if the NMR spectroscopic data are described. Indeed, almost all the compounds have been previously reported, and in my opinion, it would have been better to report the relative bibliographic references (or the procedures, if different from that already described). To report only the spectroscopic data is not significant.

In paragraphs 2.3 and 2.5 it should be clarified which data are not reported and the reason for the selection of the nine derivatives studied. In paragraph 2.7 a deeper discussion of the results obtained with docking studies would be appropriate.

Finally, the manuscript must be carefully checked for English language.

Author Response

We uploaded response to reviewer's comments

Reviewer 2 Report

Scheme 1   is too short and not informative enough , It has to explain what are R and R’ and what are the reactions conditions , For one who is not in the field it is not clear what are the cyclisation and reduction conditions. Also the specific conditions leading to the 12 derivatives have to be given.50 L of velutin  - the sign  before L  , here and in the continuation is not clear

line  319 & 324  ; Fig. 5  is not a synthesis scheme  but rather a figure od structures

Author Response

We uploaded the file containing response to reviewer's comments

Round 2

Reviewer 1 Report

The synthesis of tested compounds has now been addedd to the manuscript, even if the fully procedures are still lacking. Anyway, some of the compounds have been already reported elsewere and the NMR spectra can help to better elucidate the structures. For these reasons, in my opinion the paper is now acceptable for publication

Reviewer 2 Report

The paper now is suitable for publication.